

# Using geochemistry to understand the sources and mean transit times of stream water in an intermittent river system: the upper Wimmera River, southeast Australia

Zibo Zhou[1], Ian Cartwright[1], Uwe Morgenstern[2]

[1]School of Earth, Atmosphere and Environment, Monash University, Clayton, VIC, 3800, Australia

[2]GNS Science, Lower Hutt, 5040, New Zealand

*Corresponding to:* Zibo Zhou (email: zibo.zhou@monash.edu)





## Abstract

Determining the mean transit times (MTTs) and water sources in catchments at different flow conditions helps better understand river functioning, manage riverine system health and water resources, and discern the responses to climate change and global water stress. Despite being common in a range of environments, understanding of MTTs and variable water sources in intermittent streams remain incomplete compared to perennial streams. Major ion geochemistry,

stable isotopes, $^{14}C$, and $^3H$ were used in this study to identify water sources and MTTs of a periodically-intermittent river from southeast Australia at different flow conditions, including zero-flow periods. The disconnected pool waters during the zero-flow period in the summer months of 2019 had $^3H$ activities of 0.64 to 3.29 TU. These and the variations in total dissolved solids and stable isotopes imply that these pools contained a mixture of groundwater and younger

evaporated stream water. $^3H$ activities during the high-flow period in July 2019 were 1.85 to 3.00 TU, yielding MTTs of up to 17 years. The $^3H$ activities at moderate and low-flow conditions in September and November 2019 ranged from 2.26 to 2.88 TU, implying MTTs of 1.6 to 7.8 years. Regional groundwater near the Wimmera River has $^3H$ activities of < 0.02 to 0.45 TU and $^{14}C$ activities of 57 to 103 pMC and is not recharged by the river at high flows. The Wimmera River

and other intermittent streams in southeast Australia are sustained by smaller volumes of younger catchment waters than comparable perennials streams, indicating that near-river stores have significant impacts on maintaining streamflow during low-flow periods than older deeper regional groundwater. These smaller reservoirs result in the intermittent streams being more susceptible to changes of climate and streamflow and necessitate protection of near-river corridors to maintain

the health of the riverine systems.



## 1 Introduction

Understanding the timescale of water flow through catchments to rivers at different hydrological conditions is vital for effective water resources management, protecting riverine system, and

predicting the changes in river functioning due to climate variability, changes in land use and water utilization (Sophocleous, 2002; Cook., 2013; Van Dijk et al., 2013; Gleeson et al., 2016; Segura et al., 2019). Mean transit times (MTTs) represent the average time for precipitation to be transmitted from a recharge area through a catchment to where it discharges into rivers or streams (Cook and Bohlke, 2000; McDonnell et al, 2010; Morgenstern et al., 2010). While transit time

distributions provide better information on catchment processes than MTTs (McDonnell et al, 2010), MTTs are important for understanding how the water stores that discharge to streams vary at different flow conditions (McGuire and McDonnell, 2006; Blavoux et al., 2013; Duvert et al., 2016; Howcroft et al., 2018). Streams with long MTTs may be sustained by larger volumes of water from within the catchments (e.g., Morgenstern et al., 2010; Gusyev et al., 2016; Howcroft

et al., 2018) and be less sensitive to short-term climate variability (e.g., drought lasting years to decades). Thus, MTTs are important for predicting the resilience of catchments. In addition, the MTTs may control water salinity, water temperature, microbial activity, and the attenuation and dispersion input of nutrients and other contaminants to rivers (Kirchner et al., 2000; Hare et al., 2021).

Rivers may be sustained by inflows of water that range from a few days to several centuries old. Younger water stores may be derived from stores in the shallow near-river environment (such as surface runoff, water stored in the soil, and interflow), while regional groundwater is a large volume store of older water (Soulsby et al., 2000; McGuire and McDonnell, 2006; Stewart et al., 2010; Cartwright and Morgenstern, 2015; Duvert et al., 2016; Jung et al., 2019). However,

relatively little is known about the timescale of water flow in most catchments and whether water

of different ages contributes to river flow at different flow conditions. Numerous studies have

focused on perennial streams and have revealed the presence of long-lived water stores

contributing to streamflow especially during low-flow periods (Rice and Hornberger, 1998;

Soulsby et al., 2006; Hrachowitz et al., 2009; Cartwright and Morgenstern, 2015; Gusyev et al.,

2016; Howcroft et al., 2018; Cartwright et al., 2020). There has been less attention on intermittent

streams, which represent > 50% of global rivers and are especially important in semi-arid areas

(Datry et al., 2014; Costigan et al., 2015; Gutiérrez-Jurado et al., 2019, Shanafield et al., 2021).

The connection between intermittent streams and regional groundwater may be less important than

for perennial streams, especially during the periodic cease-to-flow times when the water table falls

and water from near-river stores become dominant (e.g., Zimmer and McGlynn, 2017).

Determining MTTs of intermittent streams will improve our understanding of surface water and

groundwater interaction and allow us to better recognize the importance of young water in

intermittent streams.

1.1 Documenting mean transit times

Several approaches can be used to estimate MTTs in rivers. MTTs may be determined by using

lumped parameter models (LPMs) that describe the distribution of water with different ages or

tracer concentrations in homogeneous aquifers with simple geometries and consistent recharge

rates (Maloszewski and Zuber, 1982; Maloszewski, 2000; Zuber et al., 2004; McGuire and

McDonnell, 2006).

LPMs may be used to estimate MTTs based on the attenuation of $\delta^{18}O$ values or Cl concentration

variabilities in rainfall at the catchment outlet. This approach requires sub-weekly measurements

of tracer concentrations in rainfall and stream water, and such datasets are available only in a small



number of catchments globally. Because intermittent streams only flow for part of the year, it is more difficult to use LPMs based on continuous $^{18}$O or Cl measurements than in perennial streams.

In addition, this approach assumes that the catchment is at steady state, which is unlikely to be the case (Kirchner, 2016b). It is also not viable where MTTs are greater than 4 to 5 years due to attenuation of the input record to below the resolution at which the tracers can be measured (Stewart et al. 2010), which is commonly the case in southeast Australia (Cartwright et al., 2020). Techniques such as ensemble hydrographs (Kirchner, 2019; Knapp et al., 2019), flux tracking

(Hrachowitz et al., 2013), and StoreAge Selection Functions (Rinaldo et al., 2015) can determine transit times from shorter time-series and do not assume steady-state conditions. However, these methods still require intensive measurements (such as sub-weekly tracer data for rainfall and streams) that are not commonly available.

Tritium ($^3$H) has a half-life of 12.32 years and is part of the water molecule. Unlike tracers such as

chlorofluorocarbons, and SF$_6$, $^{14}$C, and $^3$He, its abundance is not affected by degassing or geochemical or biogeochemical reactions. This allows $^3$H to be used to estimate MTTs of shallow groundwater, water from the unsaturated zone and stream water (e.g., Morgenstern et al., 2010; Duvert et al., 2016; Jung et al., 2019). Due to atmospheric nuclear tests, $^3$H activities in rainfall reached a peak in the 1950s and 1960s (the "bomb pulse"). In the southern hemisphere, the remnant

bomb pulse $^3$H activities are now lower than those in modern rainfall (Morgenstern et al., 2010; Tadros et al., 2014). This makes it possible to estimate MTTs from a single $^3$H measurement in a similar way to other radioisotopes such as $^{14}$C and $^{36}$Cl are used to determine residence times of old groundwater (e.g., Clark, 2015; Cartwright et al., 2017; Howcroft et al., 2019). Low-level $^3$H measurements allow MTTs of up to ~150 years to be determined, although the relative precision

of the estimates decreases at longer MTTs.





MTT estimates made using $^3$H have several uncertainties. The decline of the bomb pulse $^3$H activities in the southern hemisphere makes it impossible to assess the suitability of an LPM by time-series $^3$H measurements that commence now (Cartwright and Morgenstern, 2015). Assigning LPMs is therefore based on catchment attributes (e.g., the geometry of the flow system) and

information from previous studies in similar catchments. Although it represents an uncertainty, MTTs are less sensitive to the choice of LPMs than in the northern hemisphere (Morgenstern et al., 2010; Blavoux et al., 2013). In addition, where multiple water sources (e.g., groundwater, soil water, or water from multiple tributaries) with different MTTs contribute to rivers (aggregation), it is difficult to estimate MTTs (Suckow, 2014; Kirchner, 2016a; Stewart et al., 2017). The

heterogeneous hydraulic conductivities of aquifers also contribute uncertainty to MTT calculations (Weissmann et al., 2002; McCallum et al., 2015). However, where the scale of heterogeneity is small relative to the size of the aquifer, the MTTs are similar to those predicted by the LPMs (Cartwright et al., 2018). Lastly, the seasonal variability of $^3$H activities in rainfall could lead to uncertainty in MTT estimates. Where strong seasonal recharge occurs, $^3$H activities of rainfall that

recharges the catchment may be different from those of annual rainfall, which is usually used as the $^3$H input (e.g., Morgenstern et al., 2010). Although these factors introduce uncertainties in MTT calculations, water in the southern hemisphere with lower $^3$H activities invariably has longer MTTs, which allows relative relationships to be determined. This also permits the understanding of the changing sources of water contributing to streams during different flow conditions (Duvert et al.,

2016; Howcroft et al., 2018; Cartwright et al., 2018, 2020).

Previous studies of perennial streams in southeast Australia (summarized in Cartwright et al., 2020) noted that the runoff coefficient (the proportion of annual rainfall to be exported by the stream) had an inverse correlation with MTTs. This relationship probably reflects the high





evapotranspiration rates in some catchments which results in low recharge rates, slower

groundwater flow, and less of the rainfall being exported as runoff. The runoff coefficient

represented a more viable first-order proxy for MTTs than catchment attributes such as slope,

drainage density, or major ion concentrations in those catchments. Whether a similar relationship

between MTTs and runoff coefficient occurs in intermittent catchment is not known.

1.2 Understanding water sources

As noted above rivers are potentially fed by a range of water stores from within catchment,

including soil water, interflow, bank return flow, shallow riparian groundwater, and deeper

regional groundwater (e.g., Peters et al., 2014; Duvert et al., 2016; Cartwright and Morgenstern,

2018; Howcroft et al., 2019). Due to mineral dissolution, the breakdown of organic matter and

evapotranspiration, water stored within catchments commonly has high salinity than surface runoff

(Herczeg et al., 2001; Edmunds, 2009). Variable operation of these processes may result in

differences in major ion geochemistry between the water sources. For example, soil water may

have high nitrate or organic carbon concentrations due to breakdown of organic matter. There may

also be differences in the stable isotope geochemistry of these waters reflecting seasonal recharge,

evapotranspiration, or long-term changes to rainfall stable isotope ratios (Hughes and Crawford,

2012). Not all catchments, however, contain water stores with distinct geochemistry. This is

commonly the case in southeast Australia where high evapotranspiration rates mask the effects of

mineral dissolution (Herczeg et al., 2001; Cartwright and Morgenstern, 2015; Howcroft et al., 2018;

Barua et al., 2022). In those catchments, documenting MTTs allows the inputs of older and

younger catchment water to be assessed.


1.3 Objectives

This study determines the mean transit time and water sources at different flow conditions (including zero flows) in the seasonally-intermittent upper Wimmera River in southeast Australia using $^{3}$H, $^{14}$C, stable isotopes, and major ion geochemistry. A previous study (Zhou and Cartwright, 2021) used similar tracers to understand the locations of groundwater inflow to the river; however,

did not specifically address the timescale of water flow in the catchment or the volumes of water sustaining streamflow. We hypothesized that: 1) mean transit times in the Wimmera River are younger than in comparable perennial streams from southeast Australia; 2) younger near-river water stores (such as shallow riparian groundwater and bank return flows) are more important than regional groundwater in sustaining the river at all flow conditions; and 3) due to the river

containing alternate gaining and losing reaches, the runoff coefficient will not be a reliable indicator of mean transit times. As with many rivers globally (Shanafield et al., 2020; Messager et al., 2021), the upper Wimmera River has become more intermittent over recent years due to climate change. The results from this study are important in understanding and managing catchment behaviour in intermittent streams more generally. We also assess what tracers are useful in

distinguishing between water sources, which will help inform studies on similar catchments.

## 2 Study area

Located in southeast Australia, the Wimmera River is an intermittent river in the southern Murray-Darling Basin. The catchment has an area of approximately 24,000 km$^{2}$ and the middle and lower parts of the river are important for agriculture (Fletcher, 2015; Department of Environment, Land,

Water and Planning, 2021). In the summers of all but the wettest years, the Wimmera River ceases to flow and comprises a series of disconnected pools (Western et al., 1996). Since the 1980s, the Wimmera River has experienced a decline in streamflow and an increase in intermittency



(Department of Environment, Land, Water and Planning, 2021) especially during the time from 1996 to 2009 when southeast Australia experienced a large reduction in rainfall and streamflow

(known as Millennium Drought) (Bureau of Meteorology, 2021). The study area is located in the upper Wimmera River catchment (Fig. 1) which has an area of approximately 3000 km$^2$. Dryland pasture with remnant native eucalypt woodlands is the dominant vegetation coverage in the upper catchment (Fletcher, 2015; Department of Environment, Land, Water and Planning, 2021). There is only minor groundwater and river water use in the upper catchment (Robinson et al., 2006;

Wimmera Catchment Management Authority, 2013; Fletcher, 2015).

The upper catchment of the Wimmera River (Fig. 1) comprises a Palaeozoic basement of metamorphosed shales and schists of the St Arnaud Group, indurated sandstones of the Glenthompson and Grampians Groups, and Devonian granites (Department of Jobs, Precincts and Regions, 2021). Alluvial and lacustrine Palaeogene to Recent sediments that were deposited by

the precursors of the current rivers overlie the basement. These sediments consist of rounded gravels, coarse sands, silts and clays (Robinson et al., 2006). The dominant topography in the upstream reaches of the upper Wimmera River is a broad valley while the downstream part is a flat alluvial flood plain (Robinson et al., 2006; Department of Environment, Land, Water and Planning, 2021).

There is a decreasing trend of average annual rainfall from the southeast (709 mm) to northwest (505 mm) in this area (Bureau of Meteorology, 2021), and the winter and spring months (May to August) are the wettest. Similar to rainfall, high river flows occur in the winter and spring (Department of Environment, Land, Water and Planning, 2021).

Groundwater in the upper Wimmera recharges on the margins of catchment and flows northwards,

with flow paths converging on the river (Radke and Howard, 2007; Lamontagne et al., 2014: Fig.



1). River water geochemistry demonstrates that groundwater discharge locally occurs in the upper

and middle reaches of the upper Wimmera River driven by relatively steep topography and high

hydraulic gradients, whereas the lower reaches have lower groundwater inflows due to subdued

topography (Zhou and Cartwright, 2021). From the downstream trends in Cl and $^{222}$Rn and the

high $^3$H activities, the study concluded that near-river stores were likely important contributors to

streamflow.

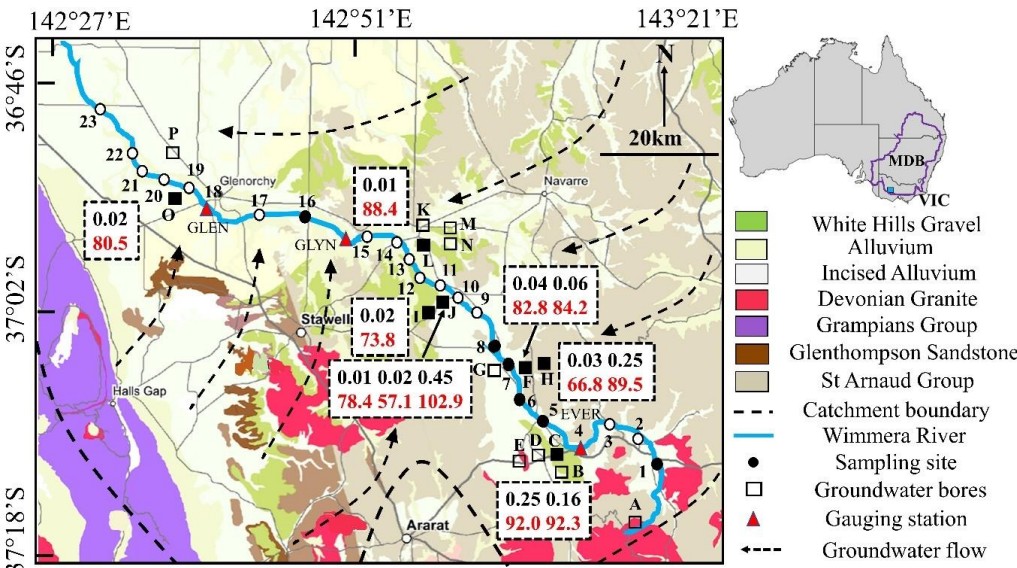

Figure 1. Summary geological and hydrogeological map of the upper Wimmera River. Stream water
sampling sites and groundwater bores are indicated by numbers and letters, respectively. The sites that have
radioactive isotope data are shown in solid symbols. Gauging stations with site number are Eversley (EVER;
415207), Glynwylln (GLYN; 415206), and Glenorchy (GLEN; 415201). Background geological map from
Department of Jobs, Precincts and Regions (2021) (© State Government of Victoria 1996-2021); other
information from Robinson et al. (2006), Department of Environment, Land, Water and Planning (2021),
and Zhou and Cartwright (2021). Boxes show $^{14}$C (red) and $^3$H activities (black) of groundwater (data from
Table 2).



## 3 Materials and methods

3.1 Sampling

Stream water samples (four rounds in total) were collected between March and November 2019 in the upper Wimmera River (Fig. 1, Table S1). River samples were collected from the centre of the river ~1m below the surface, or just above the bed where the river was shallower, using an open sample collector. 15 samples of pool water were taken using an open sample collector in March 2019 when the river consisted of disconnected pools with small flowing sections. The samples were taken from isolated pools, not the flowing reaches. 23 samples were collected in July, September, and November. July and September were high and moderate flow periods, respectively, whereas November was a low flow period just before the river ceased to flow again. 13 samples of near-river water (NRW) were taken from the top of the saturated zone within 3m of the river. Due to Covid-19, the NRW samples were collected in April 2021 but the conditions were similar to March 2019. Groundwater samples were taken in November 2019 from groundwater-monitoring bores installed on the river bank and floodplain using an impeller pump (Fig. 1, Table S2). In excess of three volumes of water were extracted prior to sampling or the bores were pumped dry and allowed to recover.

There are three gauging stations along the river that continuously measure streamflow including Eversley, Glynwylln and Glenorchy on Fig. 1. (Department of Environment, Land, Water and Planning, 2021). Linear interpolation and extrapolation were used to estimate intermediate streamflow data. Runoff coefficients (the percentage of rainfall that is exported annually by the stream) were estimated using 1993 to 2021 streamflow records from three gauges (Department of Environment, Land, Water and Planning, 2021) and the average annual rainfall (Bureau of Meteorology, 2021).





3.2 Analytical techniques

$^3$H activities were analysed at the Institute of Geological and Nuclear Sciences (GNS) in New

Zealand by liquid scintillation in Quantulus ultra-low-level counters following vacuum distillation

and electrolytic enrichment (Morgenstern and Taylor, 2009). $^3$H activities are expressed in Tritium

Units (TU) and the detection limit is 0.02 TU with relative uncertainties ($1\sigma$) of approximately

$\pm 2\%$. $^{14}$C activities of groundwater were measured at GNS in New Zealand by accelerator mass

spectrometer (AMS). $CO_2$ in groundwater was extracted using orthophosphoric acid and then

converted into a graphite target after being purified under vacuum. $^{14}$C activities are expressed as

percent modern carbon (pMC), where the $^{14}$C activity of modern carbon is 95% of the $^{14}$C activity

of the NBS oxalic acid standard in 1950 (Stewart et al., 2004).

EC values were measured in the field using a calibrated TPS meter and electrode. A ThermoFischer

quadrupole ICP-OES at Monash University was used to measure cation concentrations on samples

that were filtered through 0.45μm cellulose nitrate filters and acidified to pH<2. Anion

concentrations were measured using a Thermo Fischer ion chromatograph at Monash University

on filtered, unacidified samples. Based on replicate analyses, the precision ($\sigma$) of major ion

concentrations ranges from 2 to 5 %. Stable isotope ratios were analysed at Monash University

using a ThermoFinnigan Delta Plus Advantage mass spectrometer. $\delta^{18}$O values of water were

measured in a ThermoFinnigan Gas Bench by equilibration with He-$CO_2$ at 32°C for 24-48 hours.

$\delta^2$H values of water were measured following reduction by Cr at 850°C in a Finnigan MAT

H/Device. $\delta^{18}$O and $\delta^2$H values were normalised following Coplen (1988) and are expressed

relative to V-SMOW. Precision ($\sigma$) based on replicate analyses is 0.15‰ for $\delta^{18}$O and 1‰ for $\delta^2$H.

The geochemistry data is in the Supplement.



### 3.3 Mean transit times

MTTs were calculated from single measurements of [3]H using lumped parameter model (LPMs) implemented in the TracerLPM Excel workbook (Jurgens et al., 2012). The [3]H activity of stream water at time t ($C_0$ (t)) is related to the input of [3]H ($C_i$) in rainfall overtime via the convolution

integral:

$$C_0(t) = \int_0^\infty C_i (t\text{-}\tau) g(t) e^{-\lambda\tau} d\tau \tag{1}$$

In equation (1), $\tau$ is the transit time, $t\text{-}\tau$ is the time that the water was recharged, $\lambda$ is the decay constant of [3]H (0.0563 yr [-1]), and g(t) is a function that describes the distribution of flow paths and transit times in the flow system. The [3]H input was from the annual weighted average [3]H activities

of rainfall in Melbourne (Tadros et al., 2014). Modern rainfall in central Victoria is predicted to have average annual [3]H activities in the range of 2.8 to 3.2 TU (Tadros et al., 2014); measured [3]H activities of rainfall in Victoria are within the range of the Tadros et al. (2014) estimates (Cartwright and Morgenstern, 2015; Cartwright et al., 2018; Howcroft et al., 2018; Barua et al., 2022).

Several LPMs were used. The dispersion model (DM) stems from the one-dimensional advection-dispersion transport equation and can be applied to a variety of aquifer configurations (Zuber and Maloszewski, 2001; McGuire and McDonnell, 2006; Jurgens et al., 2012). The dispersion parameter (DP), which is the ratio of dispersion to advection, needs to be defined when using this model. For kilometre-scale flow systems such as in this study, DP values of 0.05 to 0.5 are suitable

(Maloszewski, 2000; Zuber and Maloszewski, 2001). The Exponential Mixing Model (EMM) represents a flow system with a simple exponential distribution of flow paths. The Exponential Piston Flow Model (EPM) describes flow systems that have both exponential and piston-flow sections. It is an appropriate model for unconfined aquifers with vertical recharge through the



unsaturated zone and exponential flow in the saturated zone (Maloszewski, 2000; Morgenstern et

al., 2010; Howcroft et al., 2019). The EPM ratio is the relative contribution of piston to exponential

flow and values of 0.33 and 1, which represent 75% and 50% exponential flow were used. MTTs

were estimated by matching the [3]H activities predicted by the LPMs to the measured [3]H activities.

While the estimates of MTTs are based on single samples, these LPMs have successfully

reproduced long-term time-series [3]H activities of stream water in other regions (Maloszewski and

Zuber, 1982; Blavoux et al., 2013; Morgenstern et al., 2015).

The same lumped parameter models were used to calculate the predicted [14]C vs. [3]H trends and

MTTs of groundwater. For [14]C, the input function is based on activities of atmospheric $CO_2$

(McCormac et al., 2004; Reimer et al., 2013). Calcite in these aquifers is mainly cements and veins,

which are likely to have variable $\delta^{13}C$ values (Cartwright and Morgenstern, 2012; Cartwright et

al., 2013; Clark, 2015; Meredith et al., 2016), precluding the use of isotope mass-balance to

estimate the degree of closed-system calcite dissolution. However, the aquifers are siliceous and

close-system calcite dissolution is expected to be minor (< 10%) (Clark, 2015).

3.4 Volumes of groundwater

The groundwater volumes (V in $m^3$) that contribute to the river are related to MTT and streamflow

($Q$ in $m^3$ $yr^{-1}$) via:

$$V = Q \times MTT \tag{2}$$

(Maloszewski and Zuber, 1982, 1992; Morgenstern et al., 2010).

## 4 Results

### 4.1 Streamflow

The variations of streamflow at the three gauging stations (Fig. 1) along the upper Wimmera River

in 2019 are shown in Fig. 2. Although streamflow of up to 50 $m^3$ $day^{-1}$ was recorded at Glynwylln





during the summer months (January to March), the river largely consisted of disconnected pools with only minor flowing sections at this time and all samples were collected from these pools. Based on the streamflow at this and the Eversley and Glenorchy gauges (Fig. 2), continuous

streamflow is estimated to have commenced in early to mid-April. There was a significant increase in streamflow from June and it reached a peak in August (up to $6.1 \times 10^5$ m$^3$ day$^{-1}$ at Glenorchy). Streamflow then decreased over spring and summer and the river ceased to flow continuously in late November. Runoff coefficients are 6.1% at Eversley, 4.5% at Glynwylln, and 4.2% at Glenorchy.

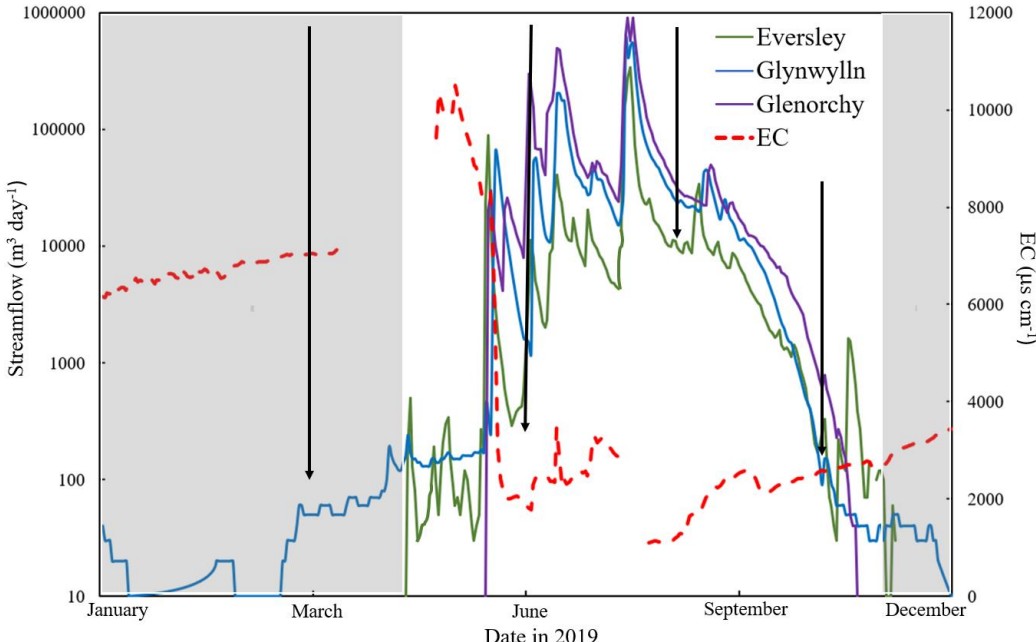


Figure 2. Variations in EC at Glynwylln (missing data are caused by measurement errors of equipment) and streamflow at Eversley, Glynwylln, and Glenorchy (Fig.1) in the upper Wimmera River in 2019. Sampling times are indicated by arrowed lines. Shaded areas represent the zero-flow periods. Data from Department of Environment, Land, Water and Planning (2021).






### 4.2 EC and major ion geochemistry

EC values in 2019 at the Glynwylln gauging station increased during the summer months, peaked at 10,500 $\mu$S cm$^{-1}$ in late May, and decreased to 1780 $\mu$S cm$^{-1}$ in June (Fig. 2). Overall, EC values were broadly inversely correlated with streamflow. TDS concentrations of stream water in the

upper Wimmera River varied from 360 to 2490 mg L$^{-1}$ (Table S1) and were also higher during the low flow period in November 2019. Na is the most abundant cation in the stream water (74-83% on a molar basis) with lower abundances of Ca (3-7%), Mg (12-19%), and K (1-2%). Cl is the most common anion (82-99% on a molar basis) in the stream water (Fig. 3). During the zero-flow period in March 2019, the stream water was much more saline with EC values of 2430-15,330 $\mu$S

cm$^{-1}$ and TDS concentrations (up to 11,420 mg L$^{-1}$) (Table S1). Near-river water (NRW) from the zero-flow period in 2021 had EC values of 1035 to 6080 $\mu$S cm$^{-1}$. TDS concentrations of regional groundwater from monitoring bores in this region ranged from 550 to 13,720 mg L$^{-1}$ (mean = 4900 $\pm$ 3770 mg L$^{-1}$: Table S2) and there is no correlation between TDS and depth. Na and Cl are again the dominant cations and anions in the groundwater (Fig. 3, Table S2). Overall, the major ion

geochemistry of the groundwater, stream water from the different flow conditions, pool water and, NRW are similar (Fig. 3).

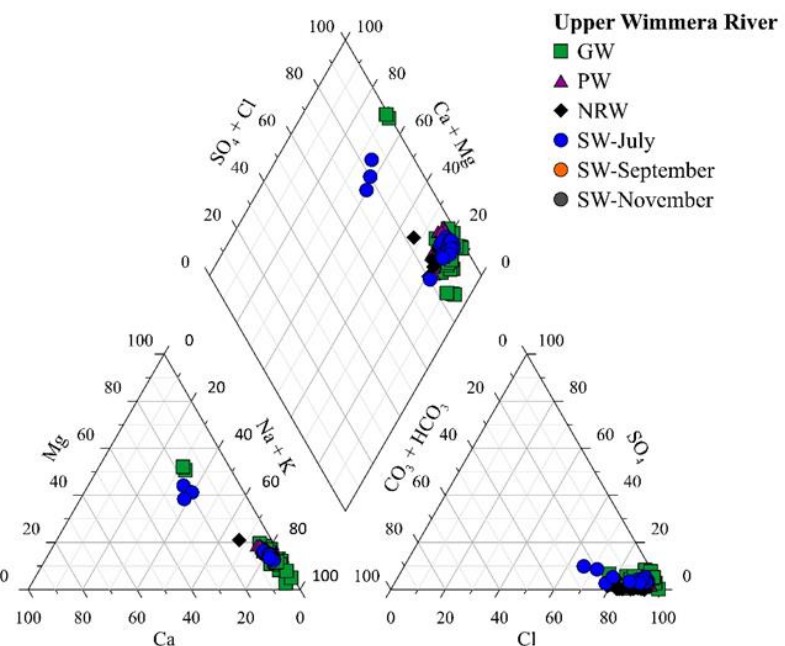

Figure 3. Trilinear diagram summarising molar major ion ratios of the different water sources in the upper Wimmera River (data from Table S1 and S2). GW=groundwater; PW= Pool Water; NRW=Near River Water; SW=Stream Water.

### 4.3 Stable isotopes

The $\delta^{18}O$ and $\delta^2H$ values of the stream water differed between the sampling rounds (Tables S1 and S2, Fig. 4). The $\delta^2H$ and $\delta^{18}O$ values of the pool water were -24 ‰ to +37 ‰ and -3.3 ‰ to +10.3 ‰, respectively and define an array to the right of Melbourne meteoric water line with a slope of 4.2 which implies that evaporation has occurred (Gonfiantini, 1986; Clark and Fritz, 1997). The pool water array intercepts the Melbourne meteoric water line at lower $\delta^{18}O$ and $\delta^2H$ values (-5.6 ‰ and -31 ‰) than those of average rainfall in Melbourne ($\delta^{18}O$ = -4.98 ‰, $\delta^2H$ = -28.4 ‰; Hollins et al., 2018), probably due to the Wimmera region being further inland. $\delta^{18}O$ and $\delta^2H$ values of river water in July and September cluster close to the Melbourne meteoric water line. $\delta^{18}O$ and $\delta^2H$ values in July 2019 were -3.6 to -7.2 ‰ (mean= -6.1 ± 0.7 ‰) and -29 to -43 ‰ (mean= -35 ± 3.8 ‰), respectively, whereas in September 2019, $\delta^{18}O$ and $\delta^2H$ values were -3.9 to





-6.1 ‰ (mean= -5.3 ± 0.4 ‰) and -29 to -32 ‰ (mean= -31 ± 0.9 ‰), respectively (Table S1).

$\delta^{18}$O and $\delta^2$H values during the low streamflow period in November 2019 ranged from -2.7 to -

4.8 ‰ and -20 to -26 ‰, respectively, while those of the NRW were -0.5 to -3.1‰ and -8 to -

29 ‰, respectively. Both the November river samples and the near-river waters define similar

evaporative trends to the pool waters. Groundwater has $\delta^{18}$O and $\delta^2$H values of -4.4 to -6.7 ‰

(mean = -5.4 ± 0.6 ‰) and -28 to -38 ‰ (mean= -31 ± 2.7 ‰) respectively that overlap with those

of the stream water in July and September (Fig. 4; Table S1).

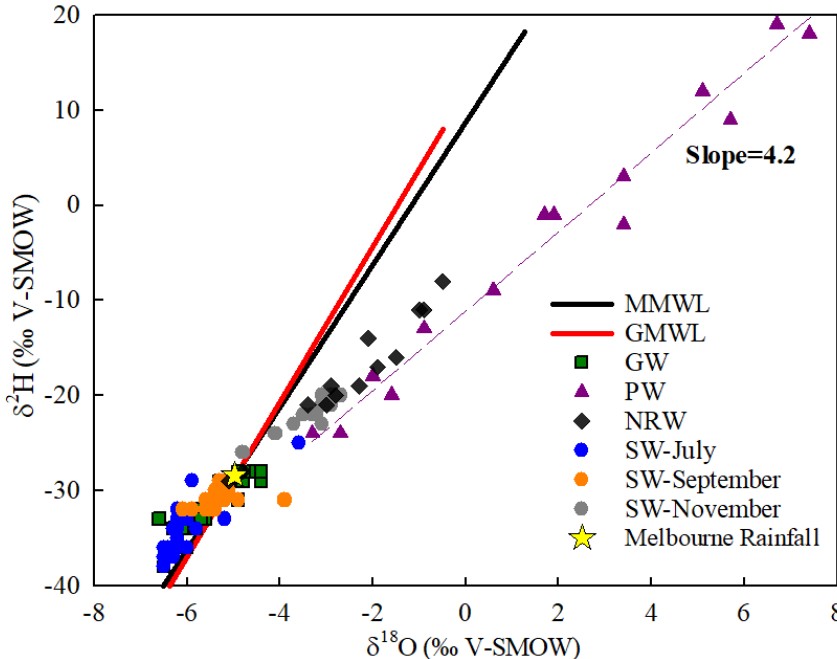


Figure 4. Stable isotope ratios of water samples in the upper Wimmera River. GMWL=Global Meteoric
Water Line ($\delta^2$H = 8.2 × $\delta^{18}$O + 11.3 ‰, as defined by Rozanski et al., 1993); MMWL= Melbourne Meteoric
Water Line ($\delta^2$H = 7.4 × $\delta^{18}$O + 8.6 ‰, as defined by Hughes and Crawford, 2012). GW=regional
groundwater; PW=pool water; NRW=Near-river water; SW=stream water. The dashed line is the best fit
for the pool water data. Data from Table S1 and S2.



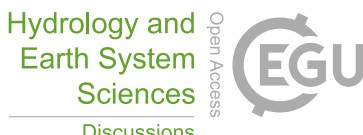

### 4.4 $^3$H and $^{14}$C activities

### 4.4.1 Regional groundwater

$^3$H activities of regional groundwater were < 0.02 to 0.45 TU (Table 2), which are significantly lower than the predicted average $^3$H activity of annual modern rainfall in this area (3.0 ± 0.2 TU: Tadros et al., 2014). The higher $^3$H activities were from shallow groundwater (<18 m depth) in the upper and middle catchment, whereas deep groundwater (>30 m depth) had $^3$H activities <0.02 TU. Groundwater close to the river does not generally have high $^3$H activities (Fig. 1). $^{14}$C activities of regional groundwater ranged between 57.1 and 103 pMC (Table 2). The highest $^{14}$C activities (up to 103 pMC) are again from the shallow groundwater in the upper and middle catchment. The trend of $^3$H vs. $^{14}$C activities (Fig. 5) are similar to those predicted for an aquifer system that does not show mixing between shallow younger groundwater and deeper older groundwater (i.e., where the activities of the two radioisotopes are controlled by their input functions and decay rates).

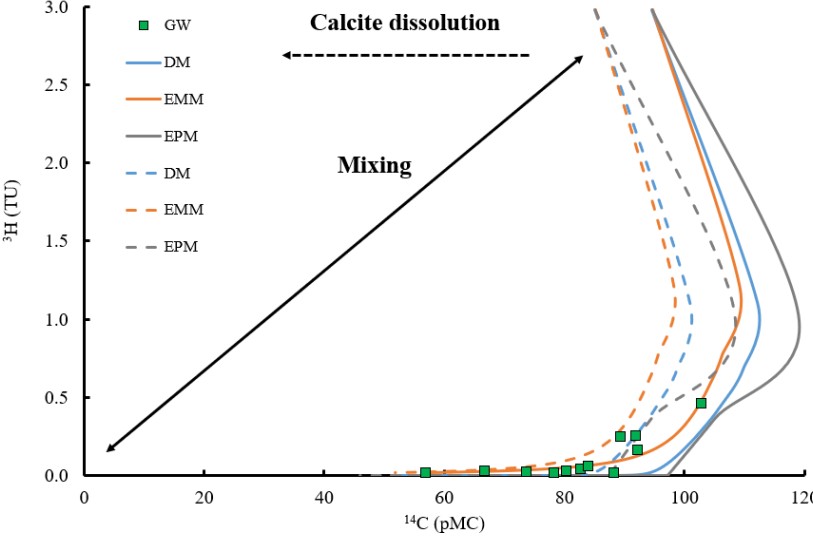

Figure 5. $^3$H activities vs. $^{14}$C activities of groundwater from the upper Wimmera River. Curved lines are the predicted covariance in the radioisotopes predicted by the Exponential Mixing, Exponential-Piston (EPM ratio = 1), and Dispersion (DP=0.5) lumped parameter models. Solid arrowed lines show schematically the effects of mixing between old regional groundwater (low $^{14}$C and $^3$H free) and modern or





recently recharged water (high $^{14}$C and $^{3}$H). Calcite dissolution lowers the predicted $^{14}$C activities (dashed lines are used to display 10% calcite dissolution).


4.4.2 River water

The $^{3}$H activities of pool water varied from 0.64 to 3.29 TU (Fig. 6; Table 1). The highest $^{3}$H activity (3.29 TU), which is higher than that of average annual rainfall, was from an area of subdued topography in the lower reaches. In contrast, $^{3}$H activities were lowest (down to 0.64 TU)

where the river is located near steeper hillslopes and flows through coarse sediments. There is a strong positive correlation ($R^2 = 0.94$) between $^{3}$H activities of pool water and $\delta^2$H values (Fig. 6a). Pool waters have a wide range of $^{3}$H activities with variable TDS concentrations, ranging from 2237 to 4639 mg L$^{-1}$. The stream water was less saline with a range of TDS 433-2038 mg L$^{-1}$ (Fig. 6b).

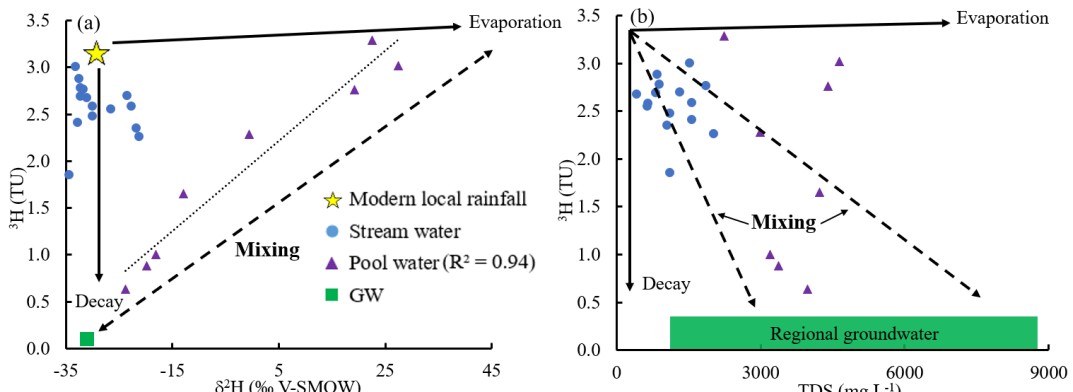


Figure 6. (a) Variation of $^{3}$H activities and $\delta^2$H values and (b) $^{3}$H activities and TDS of stream water, pool water and groundwater (GW) in the upper Wimmera catchment. arrows show trends expected from evaporation and mixing. Data from Table S1 and S2.

The $^{3}$H activities of stream water during the periods when the river was flowing were generally lower than those of rainfall and had a range of 1.85 to 3.00 TU in July, 2.48 to 2.88 TU in September, and 2.26 to 2.69 TU in November (Table 1). During the high streamflow in July, the $^{3}$H activities of river water were more variable than in September and November (Figs. 6, 7). Both lowest and highest values of $^{3}$H activities were recorded in the high flow period (Fig. 7).

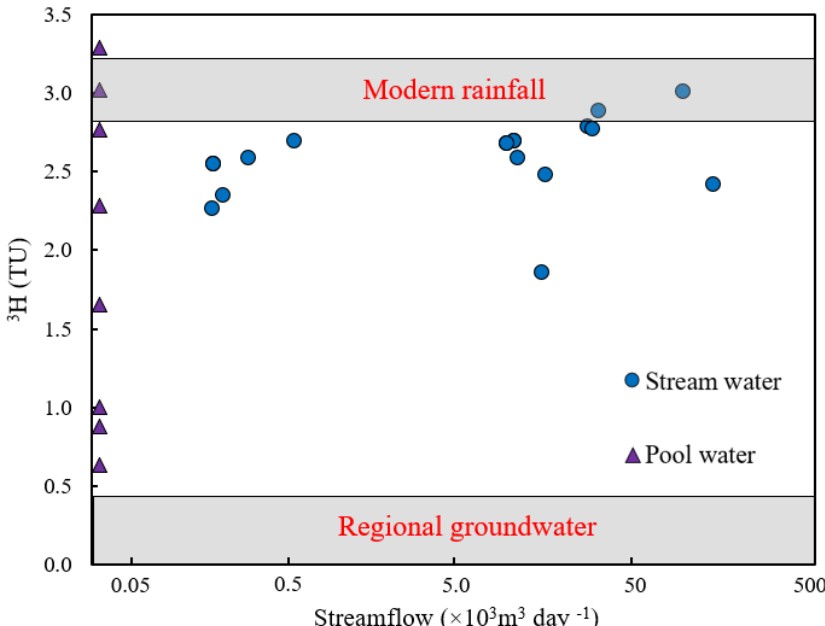

Figure 7. Variation of $^3$H activities and streamflow. Pool water represents a time of zero streamflow. Shaded areas show $^3$H range of modern rainfall and regional groundwater. Data from Table 1.

## 5 Discussion

The combined streamflow, major ion geochemistry, and stable and radioactive isotopes allow the

water sources contributing to streamflow and MTTs at different flow conditions to be understood.

5.1 Identification of water sources

Pool waters in summer months contain the last remnants of river water from when the river ceases

to flow, rainfall and/or groundwater discharging from the underlying aquifers (Cartwright and

Morgenstern, 2016; Lamontagne et al., 2021). In the upper Wimmera River, most of the pools are

perennial and have a wider range of $^3$H activities than stream water and groundwater (Fig. 7). The

variation in $^3$H activities with $\delta^2$H (Fig. 6a) and TDS concentrations (Fig. 6b) most likely reflects

the mixing between older regional groundwater and younger evaporated stream water. Some of

the pool water has higher $^3$H activities than were recorded at the low flows that immediately





precede the formation of the pools (Fig. 7), which may reflect the direct input of rainfall over the summer months into the pools. Summer rainfall in southeast Australia, however, generally has $^3$H activities close to the annual average (Tadros et al., 2014), which is problematic for explaining the locally high $^3$H activities (up to 3.29TU). Late winter and spring rainfall have higher $^3$H activities than those of average rainfall due to stratosphere-to-troposphere moisture exchange (Tadros et al.,

2014). The upper Wimmera River is locally losing at high streamflows, such as commonly occur in late winter and early spring (Fig. 2) allowing bank storage to occur. Subsequent drainage of bank water back to the river potentially explains the local high $^3$H activities in the pool. Alternatively, these high $^3$H activities may reflect the input of young water from perched aquifers in the riparian zone as documented in intermittent streams elsewhere in western Victoria (Barua et

al., 2022).

The $^3$H activities of stream water when the upper Wimmera River is flowing is much higher than that of groundwater (Figs. 6, 7), implying that the river is largely fed by young water. This is the case even during the low flow periods, which is when rivers are most likely to be sustained by long-lived water stores (e.g., Gusyev et al., 2016; Cartwright et al., 2020). The much lower TDS

concentrations of the river water compared with the groundwater (Fig. 6b) and irregulated downstream trends in major ion concentrations (Table S1) are also consistent with the input of water from mainly near-river sources. The one lower $^3$H activity (1.85 TU) during high flow conditions in July 2019 (Table 1, Fig. 7) may reflect very local input of regional groundwater or older near-river waters being flushed into the stream during the early stages of rainfall. Unlike in

some catchments (Tsujimura et al., 2007; Birks et al., 2019; Jung et al., 2019), the major ion and stable isotope geochemistry of regional groundwater and near-river water are similar (Figs. 3 and 4; Tables S1 and S2). The geochemistry of the stream also does not vary with flow. This precludes





using these tracers to distinguish water sources or as a proxy for [3]H activities (e.g., Peters et al., 2014; Cartwright and Morgenstern, 2015; Beyer et al., 2016; Cartwright et al., 2020). The large

difference in [3]H activities between regional groundwater and rainfall, however, explicitly allows the input of older groundwater to be assessed.

5.2 Mean transit times of river water

The estimates of mean transit time assume that there is a single flow system within the catchment. The pool waters probably represent discrete mixing between older groundwater and younger water

(Fig. 6). It is not possible to calculate the MTTs of these waters using a single LPM and there is insufficient data to use binary LPM calculations. In common with other studies of MTTs, it is assumed that when the river is flowing it is sustained by a single store of water with MTTs that vary as the catchment dries down and wets up. As discussed above, most of the water sustaining the river when it is flowing is likely to be derived from near-river stores with little input from

regional groundwater, which is consistent with that conceptualisation.

The MTTs in the upper Wimmera River when it was flowing ranged from < 1 to 17 years and are mostly less than 7 years (Table 1). The different LPMs yielded slightly different MTTs and the range of MTTs increases with decreasing [3]H activities. The highest estimates of MTTs are from the EMM and the lowest are from the DM with $D_p$ 0.05. During the high flow period in July 2019,

MTTs were generally higher (<1 to 16.8 years). By contrast, the range of MTTs at moderate and low flow conditions were 1.6 to 7.8 years (Table 1).

The MTTs are subject to several uncertainties. The uncertainty arising from the choice of LPMs is greater at [3]H activities <2.5 TU with an average uncertainty of 22% (Table 1). The influence of uncertainties in the [3]H activities of modern rainfall (±0.2 TU: Tadros et al., 2014) may be

demonstrated using the EPM with an EPM ratio of 0.33 (the effects are similar in other models)





(Fig. 8a). Varying the $^3$H activities between 2.8 TU and 3.2 TU translates into uncertainties of ±6 to 7% (Fig. 8a). Applying a similar 10% uncertainty to the entire $^3$H input function produces uncertainties of 9 to 21%, with the largest difference when $^3$H activities were greater than 1 TU (Fig. 8b). Uncertainties arising from the precision of the $^3$H analyses are <0.8 years. Mixing of

multiple water sources with different MTTs (aggregation) may result in actual MTTs being lower than calculated MTTs (Suckow, 2014; Kirchner, 2016a; Stewart et al., 2017). Aggregation has the most impact when there is binary mixing between water with very different MTTs. Mixing between multiple water stores with a range of MTTs has less impact on the MTTs calculated using $^3$H as that scenario is similar to what is modelled using the LPMs (Cartwright and Morgenstern,

2016). In the case of the upper Wimmera River, the smaller range of MTTs implies that aggregation may not be as significant as in other catchments in southeast Australia where the range of MTTs in the catchment waters is much larger. Considering the uncertainties from the different LPMs, the analytical uncertainty, and the tritium activities of modern and historical rainfall, the range of MTTs for a $^3$H activity of 1.5 TU was 15.2 to 25.5 years, which is a relative uncertainty

of -24 to +27%. For water with 0.5 TU, the MTTs ranged 92.6 to 108 years, which is uncertainty of -9 % to +15%. Although these are substantial, it does not alter the conclusion that the upper Wimmera River is fed by relatively young water at all stages of flow.

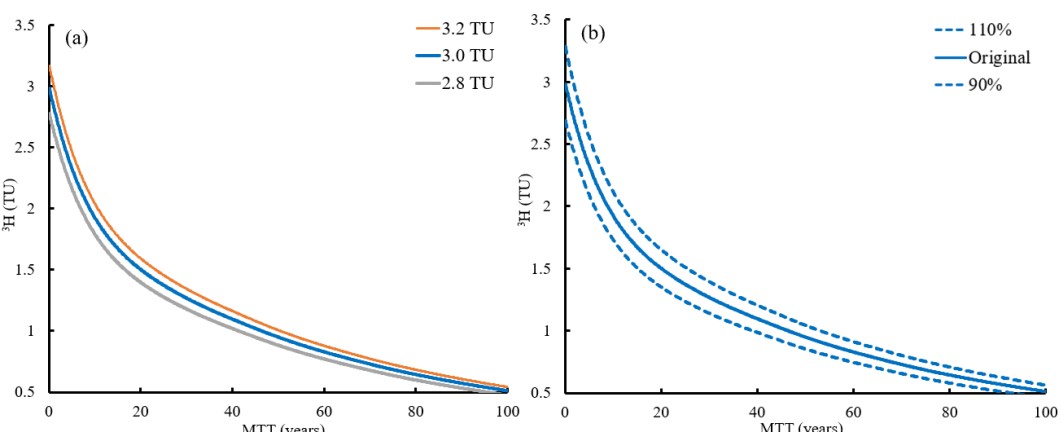

Figure 8. (a) Impacts of varying the $^3$H activity of modern rainfall from 2.8 to 3.2 TU on MTTs calculated
using the Exponential-Piston Flow Model (EPM ratio = 0.33). (b) Impacts of varying $^3$H activity of rainfall
between 90% and 110% of its assumed values on mean transit time calculated using the Exponential-Piston
Flow Model (EPM ratio = 0.33).

The average annual rainfall $^3$H activities were used in the MTTs calculations. However, if there is

strong seasonal recharge due to summer rainfall being lost by evapotranspiration, the $^3$H activities

of the recharging water may be different to the average annual $^3$H activity (Morgenstern et al.,

2010; Blavoux et al., 2013). Monthly variations in rainfall $^3$H activities are less than 1 TU and the

$^3$H activities of summer rainfall are close to annual rainfall values (e.g., Tadros et al., 2014).

Considering the general uncertainty in the $^3$H input function, uncertainties that originate from

adopting the average annual $^3$H activity are minor.

The volume of water store that sustained the streamflow calculated from the MTTs (Eq. (2)) ranged

from $3.2 \times 10^5$ m$^3$ to $2.6 \times 10^8$ m$^3$ (Fig. 9b). The estimated volume of water stored in the riparian

zone of the river is $3.1 \times 10^5$ m$^3$, which was calculated from the estimated river length (120,000

m), the width and depth of the riparian zone (6.5 m and 2 m, respectively), and an assumed porosity

of 0.2. This value is three orders of magnitude smaller than that of the calculated volume of water

needed to generate streamflow during the high flow period, implying that the streamflow was

generated from water derived from the broad landscape. By contrast, the volume of water in the





riparian zone is similar to the volume needed to generate streamflow at low flow conditions, which

indicates that it may be derived from near-river stores.

5.3 Groundwater transit times

Groundwater MTTs were calculated using the EPM model (EPM ratio = 0.33; Table 2). This model

is applicable to groundwater flow systems where the bores sample deeper groundwater flow paths

but not the short near-surface flow paths (Maloszewski and Zuber, 1982). MTTs of groundwater

with [3]H activities >0.25 TU were calculated using [3]H. For groundwater with lower [3]H activities,

the [14]C activities were used. As discussed above, the proportion of DIC from closed system calcite

dissolution in these siliceous aquifers is likely to be minor and MTTs were calculated assuming

up to 10% addition of [14]C-free carbon. The estimated MTTs of groundwater were between 120

and 5690 years (Table 2). The relative uncertainties on these estimates are likely to be similar to

those discussed above. Groundwater within a few 10s to 100s meters of the river, such as at locality

I and J in Fig. 1, had MTTs of up to 5690 years and the [14]C and [3]H activities show little evidence

of mixing (Fig. 5), implying that there is limited recharge of regional groundwater by stream water

even when the river is losing. The large contrast between the MTTs of groundwater and river water

also implies that the regional groundwater flow system is distinct from local near-river flow system.

5.4 Comparison with perennial streams

In southeast Australia, the [3]H activities at low flows in the upper Wimmera River and other

intermittent streams are higher than in perennial streams of comparable size from catchment with

similar geology and landuse (Fig. 9). Perennial streams elsewhere in Australia and New Zealand

also locally have low [3]H activities at low flows (Stewart et al., 2010; Duvert et al., 2016). This

implies that intermittent streams at low streamflows are sustained by much younger water than



perennial streams. This is most likely due to a much weaker connection between intermittent

streams and deeper older regional groundwater than is the case for perennial streams.

Because evapotranspiration rates, local vegetation types and rainfall influence both how much of

rainfall is exported via the stream and the MTTs. $^3$H activities in perennial streams from southeast

Australia correlate with the runoff coefficient (Fig. 9a). There is a broad correlation ($R^2$=0.58)

between $^3$H and runoff coefficients from multiple perennial catchments in southeast Australia

(including the Ovens, Latrobe, Gellibrand, and Yarra catchments: Fig. 9a) and the correlations in

individual catchment are higher ($R^2$ of 0.72 to 0.94: Cartwright et al., 2020). By contrast, the $^3$H

activities in the Wimmera and other intermittent streams are much higher at comparable runoff

coefficients and are poorly correlated (Fig. 9a). This may be due to the alternating gaining and

losing conditions in intermittent streams.

The volumes of the stores of water in the catchment that contributes to streamflow in the upper

Wimmera River ($3.2 \times 10^5$ m$^3$ to $2.6 \times 10^8$ m$^3$: Fig. 9b) are 1-2 orders of magnitude smaller at

similar streamflows than those in perennial streams from southeast Australia (up to $8.3 \times 10^9$ m$^3$

in the Ovens catchment) (Fig. 9b) but are similar to other intermittent streams (Deep Creek and

Gatum catchments). These differences are also due to the intermittent streams being less well-

connected to the deeper groundwater, which has larger volumes.

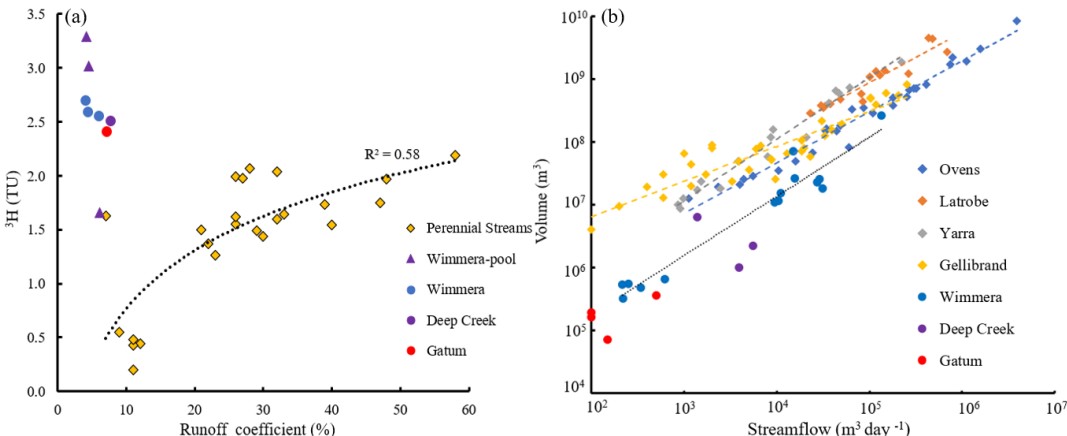

Figure 9. (a) Comparison of $^3$H activities and runoff coefficients at low and zero flow conditions and (b) volumes of water sustaining streamflows between the upper Wimmera River and other intermittent and perennial streams in southeast Australia. Perennial streams are the Ovens, Latrobe, Yarra, and Gellibrand (data from Cartwright et al., 2020); intermittent streams are Deep Creek and Gatum (data from Cartwright and Morgenstern, 2016 and Barua et al., 2022).

## 6 Conclusions

Our understanding of the functioning of groundwater-surface water interaction in intermittent catchments remain incomplete (e.g., Datry et al., 2014; Shanafield et al., 2020, 2021) and documenting the sources and transit times of water in these catchments helps address these knowledge gaps. In comparison with perennial rivers, streamflow in intermittent rivers such as the upper Wimmera River is sustained by much younger near-river water stores and these rivers may not be connected to the larger regional groundwater systems. Therefore, these intermittent rivers will be likely to more vulnerable to short-term variability of climate than comparable perennial rivers. Currently about 51 to 60% of global rives have ceased to flow at least one day per year and intermittency of streams is forecasted to increase due to climate change and rising water usage (Messager et al., 2021). This is the case for the Wimmera River where flow decreased and intermittency increased following the onset of the Millennium drought. Because intermittent streams commonly occur in semi-arid regions with scarce surface water resources, their



vulnerability to climate change potentially has significant consequences for riverine ecosystems. The progressive conversion of perennial rivers to intermittent rivers increases the number of systems that are potentially at risk.


*Data availability.* All analytical data are presented in the Supplement.

*Author contributions*. ZZ and IC conducted the sampling. ZZ and carried out the analysis of the geochemical parameters at Monash University and the MTT calculations. UM was responsible for the $^3$H and $^{14}$C analysis. All authors were involved in writing the article.

*Competing interests.* The authors declare that they have no conflict of interest.

*Acknowledgements.* This work was supported by ARC Special Research Initiative 0800001 and

Monash University. Dr. Massimo Raveggi and Rachelle Pierson are thanked for help with the analyses at Monash University.






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





# Tables

**Table 1**. $^3$H activities and calculated mean transit times (MTTs) from pool water and stream water of upper Wimmera River

| Site number[1] | Sample ID | Streamflow | $^3$H | MTTs (years) | | | | |
|---|---|---|---|---|---|---|---|---|
| | | m$^3$ day$^{-1}$ | TU | DM (0.05) | DM (0.5) | EPM (0.33) | EPM (1.0) | EMM |
| | *March 2019* | | | | | | | |
| 1 | Elmhurst | 0 | 1.65 | nc[2] | nc | nc | nc | nc |
| 4 | Ever 1 | 0 | 0.88 | nc | nc | nc | nc | nc |
| 5 | CE1 | 0 | 1.00 | nc | nc | nc | nc | nc |
| 6 | CE2 | 0 | 2.28 | nc | nc | nc | nc | nc |
| 7 | Joel 1 | 0 | 2.76 | nc | nc | nc | nc | nc |
| 8 | Joel 2 | 0 | 0.64 | nc | nc | nc | nc | nc |
| 16 | Campbell | 0 | 3.02 | nc | nc | nc | nc | nc |
| 18 | Glenorchy | 0 | 3.29 | nc | nc | nc | nc | nc |
| | *July 2019* | | | | | | | |
| 1 | Elmhurst | 10500 | 2.69 | 2.8 | 3.1 | 3.0 | 2.9 | 3.1 |
| 5 | CE1 | 14919 | 1.85 | 10.9 | 14.9 | 13.0 | 11.3 | 16.8 |
| 8 | Joel 2 | 28949 | 2.76 | 2.4 | 2.5 | 2.4 | 2.4 | 2.5 |
| 16 | Campbell | 91617 | 3.00 | 0.1 | 0.1 | 0.1 | 0.1 | 0.1 |
| 18 | Glenorchy | 135000 | 2.41 | 4.8 | 5.7 | 5.3 | 5.0 | 5.8 |
| | *September 2019* | | | | | | | |
| 1 | Elmhurst | 9500 | 2.67 | 3.0 | 3.3 | 3.1 | 3.0 | 3.3 |
| 5 | CE1 | 10898 | 2.58 | 3.6 | 4.0 | 3.8 | 3.6 | 4.0 |
| 8 | Joel 2 | 15698 | 2.48 | 4.3 | 5.0 | 4.6 | 4.4 | 5.1 |
| 16 | Campbell | 27077 | 2.78 | 2.2 | 2.4 | 2.3 | 2.2 | 2.4 |
| 18 | Glenorchy | 31110 | 2.88 | 1.6 | 1.7 | 1.6 | 1.6 | 1.7 |
| | *November 2019* | | | | | | | |
| 1 | Elmhurst | 220 | 2.55 | 3.8 | 4.3 | 4.0 | 3.9 | 4.3 |
| 5 | CE1 | 250 | 2.34 | 5.4 | 6.5 | 6.0 | 5.6 | 6.7 |
| 8 | Joel 2 | 215 | 2.26 | 6.0 | 7.6 | 6.9 | 6.3 | 7.8 |
| 16 | Campbell | 343 | 2.58 | 3.6 | 4.0 | 3.8 | 3.6 | 4.0 |
| 18 | Glenorchy | 620 | 2.69 | 2.8 | 3.1 | 2.9 | 2.8 | 3.1 |


1. Sites on Fig. 1.
2. nc = not calculated





**Table 2.** [3]H, [14]C activities, and calculated MTTs by EPM (ratio=0.33) of groundwater from the upper
Wimmera River

| Site letter[1] | Bore ID | Depth | [3]H | [14]C | MTTs-[3]H | MTTs-[14]C | MTTs-[14]C[2] |
|---|---|---|---|---|---|---|---|
| | | m | TU | pMC | years | years | years |
| C | 5242 | 23 | 0.16 | 92.3 | 197 | nc[3] | nc |
| | 5243 | 18 | 0.25 | 92.0 | 176 | nc | nc |
| F | 5227 | 28 | 0.04 | 82.8 | nc | 1591 | 691 |
| | 5228 | 14 | 0.06 | 84.2 | nc | 1436 | 551 |
| H | 5229 | 37 | 0.03 | 66.8 | nc | 3826 | 2671 |
| | 5230 | 14 | 0.25 | 89.5 | 176 | nc | nc |
| I | 5232 | 32 | 0.02 | 73.8 | nc | 2726 | 1681 |
| J | 5234 | 43 | 0.02 | 57.1 | nc | 5691 | 4421 |
| | 5235 | 30 | bd[4] | 78.4 | nc | 2111 | 1141 |
| | 5236 | 12 | 0.45 | 102.9 | 121 | nc | nc |
| L | 5379 | 7 | bd | 88.4 | nc | 1016 | 221 |
| | 2542 | 17 | 0.02 | 80.5 | nc | 1851 | 921 |

1. Sites on Fig. 1
2. MTTs estimated by [14]C activities with 10% calcite dissolutions
3. nc= not calculated