# Peer review of "Sources and mean transit times of stream water in an intermittent river system: the upper Wimmera River, southeast Australia"

_Hydrology and Earth System Sciences, 2022_

## Author Comment (AC1)

We appreciate the constructive comments from the reviewer #1 on the manuscript. The responses and proposed modifications are outlined below.

**General comment**

*The topic presented in the manuscript is relevant to HESS. The manuscript does not present novel approaches or ideas. However, it contributes to knowledge on intermittent streams by documenting sources and mean transit times of one such stream in southeastern Australia and highlighting the role of the near-river store.*

Response: Geochemical techniques are well-established methods in understanding river processes and functioning globally. However, previous studies have mainly focused on perennial streams, and studies on intermittent streams are less common. Moreover, estimates of mean transit times in rivers (especially intermittent ones) are also not overly common. The application of geochemistry, together with tritium, in intermittent streams allows us holistically understand this type of riverine systems in general. We did note those facts in the study (Section 1) and will ensure that they are highlighted in the revised version.

**Specific comments**

*1. A methodological flaw is related to the fact that the near-river samples were collected two years after the streamflow samples (April 2021 and March-November 2019).*

Response: We realise that the water from near-river corridors would be an important component when we measured the geochemistry of pool water, stream water, and regional groundwater in 2019. We had a plan to collect more river samples and near-river water samples in early 2020. Unfortunately, the Covid-19 pandemic forced Victoria into s strict lockdown for much of 2020 and 2021 and we were unable to undertake significant fieldwork at that time. There were a few short windows in 2021 when we could go in the field and luckily, we were able to take samples of near-river water in 2021 at similar flow conditions to 2019. Although not ideal, these are informative samples and we have explained the context in the text.

*2. The approach and data are well described and the interpretation of results is included in the Discussion. I would prefer if the word "likely" is less frequent there; perhaps it could sometimes be substituted by a more appropriate "we think" or "we assume".*

Response: Agreed. We will reword sentences and reduce the frequency of 'likely'.

*3. It is not clear if the average annual rainfall (lines 235-238) used in calculation of runoff coefficients for the three river gauges was estimated specifically for the upstream area of each river gauge (and how) or if the same value was used for all river gauges. Since the annual precipitation varies from 505 to 709 mm, catchment precipitation should be calculated for each gauge specifically.*

Response: It is difficult to calculate the area weighted rainfall upstream of the individual gauges with the available rainfall data. Initially, we used a single average value of rainfall for all gauges. In the revised version we will calculate a range of runoff coefficients for the gauges based on the higher and lower rainfall values. The uncertainty on runoff coefficients estimated in this way is ~15%, which does not alter the conclusion of the study. We can add the error bars to Fig. 9a.

*4. It is interesting to me that MTTs during the high flow period were generally higher (older water?) than during the low flow period (younger water?) – lines 458-461. I would assume the opposite, is it possible to comment on it briefly in the Discussion?*

Response: Yes, this is an important and interesting part of the study. High MTTs during the high flow probably reflects that older water from the catchment flushed into the river during the early stages of rainfall by hydraulic loading. This has been documented in other Australian catchments (e.g., Tambo River: Unland et al 2015, Hydrological Processes, 29, 4817-4829) and is a common feature in many river systems, sometimes referred to Old Water Paradox. The water that contributes to low flows probably includes a component of young water stored in the riverbank from the sustained winter streamflow that drains back into the river as flows subside. We will make sure that this is discussed clearly in the discussion section.

**Other comments**

*1. Title - I propose to change the title and omit the general term "geochemistry" there. Geochemistry of major ions is not used in the interpretation of data presented in the Discussion, because "...the major ion and stable isotope geochemistry of regional groundwater and near-river water are similar ...and the geochemistry of the stream does not vary with flow" (lines 440-443). Perhaps the reason of using "geochemistry" in the title was to say that it used the tools and principles of chemistry (a generals definicition of the sciences of geochemistry). However, then the application of isotopes on which is the work heavily based, is not clear from the title. "Sources and mean transit times of stream water in an intermittent river system: the*

*upper Wimmera River, southeast Australia" or "Using isotopes to understand sources and mean transit times…" could be better titles.*

Response: Agreed. We will change the title to "Sources and mean transit times of stream water in an intermittent river system: the upper Wimmera River, southeast Australia".

*2. I do not think that it is necessary to mention climate change and global water stress in second half of the first sentence of the Abstract. The manuscript does not deal with these topics. Furthrermore, presented results are not interesting only in relation to climate change or water scarcity.*

Response: Yes, we agree. We will delete those sentences as it is not the main focus of the paper.

*3. Line 60 - please check the formulation of the sentence - the water that range from days to centuries "old" – is it a correct English?.*

Response: The sentence could be clearer. We will change it to: 'the water that sustains river flow may have residence times ranging from a few days to several centuries'.

*4. Line 86 "This approach requires sub-weekly measurements of tracer concentrations in rainfall and stream water…". Since most earlier studies used monthly data with LPMs, I would not say that subweekly data are required when using attenuation of the stable isotope signal. Please think about the reformulation of the sentence.*

Response: Yes, earlier studies used less frequent data but there is a tendency to use more frequent sampling where it is available (e.g., the intensively monitored catchments such as Plynlimon). We will amend it to 'high-frequency (generally at least monthly) and long-term tracer data'. Reviewer #2 also commented on this section and further modifications are discussed below.

*5. Line 107 please check the sentence "…in a similar way to other radioisotopes such as 14C and 36Cl THAT are used to determine residence times…".*

Response: That is correct. We will add 'that' in the sentence.

*6. lines 133-134 – I do not understand the explanation of an inverse relationship between MTT and runoff coefficient that is linked to high evaporation rates. In my understanding, a higher runoff coefficient means that more precipitation goes to runoff relatively quickly, i.e. the MTT would be shorter (as the inverse relationship suggests). Where is the influence of high evaporation rate there? If the evaporation is high, the runoff coefficient should be smaller.*

Response: The runoff coefficient refers to the fraction of rainfall that is exported annually by the stream in the catchment. Higher evapotranspiration leads to a lower runoff coefficient as more water is returned to the atmosphere. Catchments with high evapotranspiration will also have lower groundwater recharge rates, and consequently less rainfall will be exported through the catchments to the streams. The lower rate of recharge results in slower flow through the catchment and consequently longer MTTs. We will ensure that this is explained clearly in the paper.

*7. line 144 "higher" (salinity) instead of "high"?*

Response: Yes, 'higher' is right and we will correct it.

*8. It would be useful to supplement Fig. 2 by one more panel with graphs showing the variability of air temperature and precipitation in 2019.*

Response: That would be helpful and we will add those to Fig. 2.

*9. line 391 - I wonder if there is an interpretation of the good correlation between 3H activity and deuterium content presented in Fig. 6a; is anything indicated by the fact that the oldest water (the one with the lowest tritium activity that is well below that of current precipitation, i.e. 0.1-1 TU) has low deuterium content while the samples representing modern precipitation (tritum activity around 3 TU) is evaporated (deuterium as high as +25 per mil)? Lines 415-417 mention that the variations "most likely" reflect mixing. I agree that is the samples plots along a line with low slope it indicates the micxing line, but could the good correlation provide any other information.*

Response: The mixing model does explain that correlation, and is consistent with other data. Firstly, TDS vs. $^3$H implies mixing of young and old water. Secondly, the interpretation that the higher $\delta^2$H values of pool water is caused by evaporation is consistent with the $\delta^{18}$O in Fig. 4. Evaporation does produce a slightly increase in 3H activities (depicted in Fig. 6a) but not of the magnitude observed in the pool waters. The trend in Fig. 6a is strong but is based on relatively few samples and perhaps additional data would have shown a great scatter. Possibly the pool with higher groundwater inputs undergo less evaporation (conceivably they may be small through flow systems). We can discuss that briefly in the discussion.

*10. Fig. 9 shows runoff coefficients for the pools. How can be runoff coefficient of a pool which is in my understanding a stangant water body calculated?*

Response: The runoff coefficient is a catchment attribute (it is the average discharge divided by the mean rainfall). There is only one runoff coefficient for each sampling point (i.e., it is not a function of flow at the time of sampling). It is valid to use this attribute for all flow conditions including the zero flow periods. We will ensure that the definition of the runoff coefficient (Section 3.1) explains this.

---

## Author Comment (AC2)

The reviewer #2 is thanked for the constructive comments on the manuscript. The responses and proposed modifications are outlined below.

**General comments**

*The work presented by Zhou et al., is relevant to HESS. The contents are comprehensively described and provide very good insights into the rivers functioning, mean water transit times and water sources in intermittent streams in southeast Australia by using major ions and electrical conductivity. However, I suggest some minor revisions commented below.*

Response: The reviewer is thanked for acknowledging importance of our work. This was the main message we are trying to disseminate in this paper.

**Specific comments**

*Watch out the grammar and the wording of sentences. For example, is it correct "contribute uncertainty to MTT" or should it be "contribute to uncertainty in MTT" (line 120)?*

Response: Thanks for the comments. We will check the grammar and reword sentences throughout the paper (including Line 120).

*The Discussions interpret the results very well. However, in my view, there may be a little more discussion in the subsection Comparison with perennial streams on how your findings are susceptible to changes in the climate (e.g., drought resilience of watersheds, limited streamflow generation processes, and changing status of the instream water quality) since you mention this important issue both in the Abstract and in the Conclusions.*

Response: Thank you for the constructive advice. We will add more content on differences between perennial and intermittent streams. These will be 'If intermittent streams are mainly sustained by small and young water stores from near-river corridors than larger volume of regional groundwater system that is common in perennial streams, then maintaining healthy of near-river environment will be critical to protect the water resources. the smaller volumes of water sustaining streamflow also mean that intermittent streams are more susceptible to short-term variations in rainfall caused by climate changes, and the flow regime of several intermittent streams in southeast Australia (including the Wimmera) have changed over recent years. Some of this discussion is already in the paper but we will highlight it.

**Technical corrections**

*Line 26: I would name the upper Wimmera River here since you are introducing the study site, rather than later on line 33, all of a sudden;*

Response: We will add 'the upper Wimmera River' on Line 26.

*Line 50: You could explain why TTDs provide better information than MTTs, since you are mentioning this (e.g., TTDs describe all the transit times of the water parcels in the streamflow; however, MTT is a common metrics for TTDs, as it represents the mean transit time of the water leaving the catchment (McGuire and McDonnell, 2006)). Then keep going with explanations and implications of MTT, as you have already written;*

Response: We will add a little more discussion that TTDs provide more information on the distribution of water ages within the sample rather than just the mean transit time. This allows a finer-scale understanding of catchment processes (e.g., changes of water stores with flow). In practice, even with time-series data, it would probably be difficult to estimate TTDs in a system as large as the upper Wimmera because the system is likely to be heterogeneous.

*Line 60-64: what about mentioning the release of water of different ages also as a function of the catchment's wet/dry conditions?*

Response: We can certainly add those details. Again, due to heterogeneities, it would be difficult to do this in a catchment of this size as the stores of water are likely to differ spatially and there may also be differences in the timing that the stores (e.g., the soils or perched riparian groundwater) in different parts of the catchment become active.

*Line 86: following up the comment of Anonymous Referee #1, quoted below "Since most earlier studies used monthly data with LPMs, I would not say that subweekly data are required when using attenuation of the stable isotope signal", I also suggest to reformulate the sentence, and stating why you say that sub-weekly or, more generally, high frequency tracer data are commonly needed. For example, it can be said that high-frequency and long-term tracer data are generally recommended to appropriately describe fast catchment-scale hydrological behaviors and the tail of the TTDs, respectively. See:*

*1. Kirchner, J. W., Feng, X., Neal, C., and Robson, A. J.: The fine structure of water-quality dynamics: the (high-frequency) wave of the future, Hydrol. Process., 18, 1353–1359, https://doi.org/10.1002/hyp.5537, 2004,*

*2. von Freyberg, J., Studer, B., and Kirchner, J. W.: A lab in the field: high-frequency analysis of water quality and stable isotopes in stream water and precipitation, Hydrol. Earth Syst. Sci., 21, 1721–1739, https://doi.org/10.5194/hess-21-1721-2017, 2017.*

Response: This was discussed above in relation to the comments of Reviewer #1. It is true that MTTs have been estimated with less frequent data. However, as those references indicate, using more frequent data would allow for TTDs to be estimated and would also allow a better understanding of when and how different stores of water become activated. Our approach in this study to use $^3$H was that it allows MTTs to be estimated in relatively large rivers without the need for time-series $^{18}$O or Cl measurements that are not available in this catchment. We realise that this approach misses some of the details of the processes, but it does provide very valuable information on catchment functioning.

*Line 334-336: 'Overall, the major ion geochemistry of the groundwater, stream water from the different flow conditions, pool water and, NRW are similar'. What about EC? Differences in EC between stream water (2430-15,330 µS cm-1) and near-river water (1035 to 6080 µS cm-1) during zero-flow period are significant, and you could explain why.*

Response: In the upper Wimmera River, regional groundwater has the highest EC values and relatively high values of EC were also recorded in pool waters. On the contrary, stream water and near-river water (NRW) are less saline as they have contributions from fresh and young water stores. The difference in EC values between pool water and near-river water is mainly caused by evaporation in the pools. Additionally, the description in line 328 to 330 is probably confusing reader as 2430-15,330 µS cm$^{-1}$ was the range for pool water (we described it as stream water at zero flows). We will clarify this.

*Line 416-418: 'The variation in $^3$H activities with $\delta^2$H (Fig. 6a) and TDS concentrations (Fig. 6b) most likely reflects the mixing between older regional groundwater and younger evaporated stream water'. It is not clear to me why you have drawn these conclusions. Could you explain better?*

Response: This was also discussed above in relation to the comments of Reviewer #1. This explanation explains the available data (including the $\delta^{18}$O and TDS). The sample with the highest $\delta^2$H records evaporation (as implied by Fig. 4) and the samples with the lowest $^3$H look to have a component of older saline groundwater. As noted above, there may be a difference between the pools with larger groundwater contributions than those dominantly fed by surface water. We will ensure that this is clearly explained in the revised paper.

---

## Author Response (AR1)

We thank the editor and reviewer for the helpful comments. We have taken all of these into account and have also made some minor changes of our own. The responses (highlight in blue). and modifications (in bold fonts) are outlined below.

**Editor**

*Please, in particular, clarify things around the relationship between runoff coefficients and MTTs, as well as the rainfall data input used in the calculation of runoff coefficients (see review #1).*

Generally, runoff coefficients and MTTs have a broad correlation. This is because lower coefficients will lead to more rainfall being returned to the atmosphere and thus decrease groundwater recharge rates, which could consequently slow the water flow through the catchment (i.e., longer MTTs).

For rainfall data inputs, as we discussed in the replies to reviewers, it is difficult to calculate the area-weighted rainfall in the catchment. That is why we calculated a range of runoff coefficients using higher and lower rainfall values from the catchment.

**We have better clarified those in the revised version. The explanation between runoff coefficients and MTTs is described on lines 132-140. We have explained how we estimate rainfall on lines 237-240 and as discussed above we have used range of rainfall data to calculate runoff coefficients and have included uncertainties that this causes.**

*I would encourage the authors to add a few take-home messages: What have we learnt from this case study in terms of process knowledge and how transferable is this knowledge (see suggestions by reviewer #2 --> for the discussion)? Please also explain the mixing between older regional groundwater and younger evaporated stream water in more detail.*

**We have rewritten the conclusion to emphasize the more general aspects of the study. We have highlighted the importance of small volume and young water stores in intermittent riverine systems. Because intermittent streams are mainly sustained by those water stores than larger volume of regional groundwater that is common in perennial streams, then they are more vulnerable to short-term climate variability. Additionally, maintaining the health of the near-river environment will be critical to protect these streams. We have also highlighted the use of tritium in understanding water sources where major ion geochemistry and stable isotopes are similar.**

**We have also clearly explained the mixing on lines 420-428. Stable isotopes, tritium, and TDS are consistent with mixing between older regional groundwater and younger stream**

**water. We have also provided an explanation of why the pools that have high inputs of groundwater have lower degrees of evaporation.**

**Referee #1**

**General comment**

*The topic presented in the manuscript is relevant to HESS. The manuscript does not present novel approaches or ideas. However, it contributes to knowledge on intermittent streams by documenting sources and mean transit times of one such stream in southeastern Australia and highlighting the role of the near-river store.*

Geochemical techniques are well-established methods in understanding river processes and functioning globally. However, previous studies have mainly focused on perennial streams, and studies on intermittent streams are less common. Moreover, estimates of mean transit times in rivers (especially intermittent ones) are also not overly common. The application of geochemistry, together with tritium, in intermittent streams allows us holistically understand this type of riverine systems in general. We did note those facts in the study.

**We have rewritten the conclusions and this better emphasizes the general importance of using geochemistry to understand processes in intermittent streams.**

**Specific comments**

*1. A methodological flaw is related to the fact that the near-river samples were collected two years after the streamflow samples (April 2021 and March-November 2019).*

We realise that the water from near-river corridors would be an important component when we measured the geochemistry of pool water, stream water, and regional groundwater in 2019. We had a plan to collect more river samples and near-river water samples in early 2020. Unfortunately, the Covid-19 pandemic forced Victoria into s strict lockdown for much of 2020 and 2021 and we were unable to undertake significant fieldwork at that time. There were a few short windows in 2021 when we could go in the field and luckily, we were able to take samples of near-river water in 2021 at similar flow conditions to 2019. Although not ideal, these are informative samples.

**We have explained the context in the text (lines 225-228). As explained above, there was unavoidable but these samples are still important.**

*2. The approach and data are well described and the interpretation of results is included in the Discussion. I would prefer if the word "likely" is less frequent there; perhaps it could sometimes be substituted by a more appropriate "we think" or "we assume".*

Agreed.

**We have reworded sentences and reduced the frequency of 'likely' (e.g., line 206; line 466; line 525; line 537).**

*3. It is not clear if the average annual rainfall (lines 235-238) used in calculation of runoff coefficients for the three river gauges was estimated specifically for the upstream area of each river gauge (and how) or if the same value was used for all river gauges. Since the annual precipitation varies from 505 to 709 mm, catchment precipitation should be calculated for each gauge specifically.*

It is difficult to calculate the area-weighted rainfall upstream of the individual gauges with the available rainfall data. Initially, we used a single average value of rainfall for all gauges. In the revised version we calculated a range of runoff coefficients for the gauges based on the higher and lower rainfall values. The uncertainty on runoff coefficients estimated in this way is ~15%, which does not alter the conclusion of the study.

**As explained above, it was not possible to calculate area-weighted estimated rainfall upstream of gauges. We used the higher and lower annual rainfall from the catchment and calculated the uncertainty of runoff coefficients. This is explained on lines 237-240.**

*4. It is interesting to me that MTTs during the high flow period were generally higher (older water?) than during the low flow period (younger water?) – lines 458-461. I would assume the opposite, is it possible to comment on it briefly in the Discussion?*

Yes, this is an important and interesting part of the study. High MTTs during the high flow probably reflect that older water from the catchment was flushed into the river during the early stages of rainfall by hydraulic loading. This has been documented in other Australian catchments (e.g., Tambo River: Unland et al 2015, Hydrological Processes, 29, 4817-4829) and is a common feature in many river systems, sometimes referred to Old Water Paradox. The water that contributes to low flows probably includes a component of young water stored in the riverbank from the sustained winter streamflow that drains back into the river as flows subside.

**We have more clearly discussed this in the discussion section (lines 446-451).**

**Other comments**

*1. Title - I propose to change the title and omit the general term "geochemistry" there. Geochemistry of major ions is not used in the interpretation of data presented in the Discussion, because "...the major ion and stable isotope geochemistry of regional groundwater and near-river water are similar ...and the geochemistry of the stream does not vary with flow" (lines 440-443). Perhaps the reason of using "geochemistry" in the title was to say that it used the tools and principles of chemistry (a generals definicition of the sciences of geochemistry). However, then the application of isotopes on which is the work heavily based, is not clear from the title. "Sources and mean transit times of stream water in an intermittent river system: the upper Wimmera River, southeast Australia" or "Using isotopes to understand sources and mean transit times…" could be better titles.*

Agreed.

**We have changed the title to 'Sources and mean transit times of stream water in an intermittent river system: the upper Wimmera River, southeast Australia'.**

*2. I do not think that it is necessary to mention climate change and global water stress in second half of the first sentence of the Abstract. The manuscript does not deal with these topics. Furthrermore, presented results are not interesting only in relation to climate change or water scarcity.*

Yes, we agree.

**We have removed the climate change comments from the abstract.**

*3. Line 60 - please check the formulation of the sentence - the water that range from days to centuries "old" – is it a correct English?.*

The sentence could be clearer.

**We have changed the sentence (lines 61-62).**

*4. Line 86 "This approach requires sub-weekly measurements of tracer concentrations in rainfall and stream water…". Since most earlier studies used monthly data with LPMs, I would not say that subweekly data are required when using attenuation of the stable isotope signal. Please think about the reformulation of the sentence.*

Yes, earlier studies used less frequent data but there is a tendency to use more frequent sampling where it is available (e.g., the intensively monitored catchments such as Plynlimon).

**We are now saying this approach requires frequent measurements but not giving specific frequency (lines 86-89). We have refrained from trying to provide a detailed review of estimating MTTs from δ¹⁸O and Cl as that is not part of this study. It is important to provide some background to estimating MTTs by various methods but we have kept this reasonably general.**

*5. Line 107 please check the sentence "…in a similar way to other radioisotopes such as 14C and 36Cl THAT are used to determine residence times…".*

**We have changed the sentence (lines 107-109).**

*6. lines 133-134 – I do not understand the explanation of an inverse relationship between MTT and runoff coefficient that is linked to high evaporation rates. In my understanding, a higher runoff coefficient means that more precipitation goes to runoff relatively quickly, i.e. the MTT would be shorter (as the inverse relationship suggests). Where is the influence of high evaporation rate there? If the evaporation is high, the runoff coefficient should be smaller.*

The runoff coefficient refers to the fraction of rainfall that is exported annually by the stream in the catchment. Higher evapotranspiration leads to a lower runoff coefficient as more water is returned to the atmosphere. Catchments with high evapotranspiration will also have lower groundwater recharge rates, and consequently less rainfall will be exported through the catchments to the streams. The lower rate of recharge results in slower flow through the catchment and consequently longer MTTs.

**We have added the above discussion to the paper (lines 132-137).**

*7. line 144 "higher" (salinity) instead of "high"?*

**We have corrected it to 'higher' (line 146).**

*8. It would be useful to supplement Fig. 2 by one more panel with graphs showing the variability of air temperature and precipitation in 2019.*

That would be helpful.

**We have added those data to Fig. 2 and discussed them in the text.**

*9. line 391 - I wonder if there is an interpretation of the good correlation between 3H activity and deuterium content presented in Fig. 6a; is anything indicated by the fact that the oldest water (the one with the lowest tritium activity that is well below that of current precipitation, i.e. 0.1-1 TU) has low deuterium content while the samples representing modern precipitation*

*(tritum activity around 3 TU) is evaporated (deuterium as high as +25 per mil)? Lines 415-417 mention that the variations "most likely" reflect mixing. I agree that is the samples plots along a line with low slope it indicates the micxing line, but could the good correlation provide any other information.*

The mixing model does explain that correlation and is consistent with other data. Firstly, TDS vs. $^3$H implies the mixing of young and old water. Secondly, the interpretation that the higher $\delta^2$H values of pool water are caused by evaporation is consistent with the $\delta^{18}$O in Fig. 4. Evaporation does produce a slight increase in $^3$H activities (depicted in Fig. 6a) but not of the magnitude observed in the pool waters. The trend in Fig. 6a is strong but is based on relatively few samples and perhaps additional data would have shown a great scatter. Possibly the pool with higher groundwater inputs undergo less evaporation (conceivably they may be small through flow systems).

**As discussed above, we have better explained mixing and also try to explain why pools with high groundwater inputs were less evaporated (lines 420-428).**

*10. Fig. 9 shows runoff coefficients for the pools. How can be runoff coefficient of a pool which is in my understanding a stangant water body calculated?*

The runoff coefficient is a catchment attribute (it is the average discharge divided by the mean rainfall). There is only one runoff coefficient for each sampling point (i.e., it is not a function of flow at the time of sampling). It is valid to use this attribute for all flow conditions including the zero flow periods.

**We have made it clear that the runoff coefficient is a catchment attribute (lines 240 to 241).**

**Referee #2**

**General comments**

*The work presented by* Zhou et al., *is relevant to HESS. The contents are comprehensively described and provide very good insights into the rivers functioning, mean water transit times and water sources in intermittent streams in southeast Australia by using major ions and electrical conductivity. However, I suggest some minor revisions commented below.*

The reviewer is thanked for acknowledging importance of our work. This was the main message we are trying to disseminate in this paper.

**Specific comments**

*Watch out the grammar and the wording of sentences. For example, is it correct "contribute uncertainty to MTT" or should it be "contribute to uncertainty in MTT" (line 120)?*

**We have checked the grammar and reworded sentences throughout the paper.**

*The Discussions interpret the results very well. However, in my view, there may be a little more discussion in the subsection Comparison with perennial streams on how your findings are susceptible to changes in the climate (e.g., drought resilience of watersheds, limited streamflow generation processes, and changing status of the instream water quality) since you mention this important issue both in the Abstract and in the Conclusions.*

**As we explained earlier, we have rewritten the conclusions to emphasize more general results of the study (lines 563 to 593).**

**Technical corrections**

*Line 26: I would name the upper Wimmera River here since you are introducing the study site, rather than later on line 33, all of a sudden;*

**We have added 'the upper Wimmera River' on line 25.**

*Line 50: You could explain why TTDs provide better information than MTTs, since you are mentioning this (e.g., TTDs describe all the transit times of the water parcels in the streamflow; however, MTT is a common metrics for TTDs, as it represents the mean transit time of the water leaving the catchment (McGuire and McDonnell, 2006)). Then keep going with explanations and implications of MTT, as you have already written;*

**We have added more discussion on this (lines 47-51). However, as noted above, we did not provide a detailed analysis of TTD estimation as it is not something that we can do in this study.**

*Line 60-64: what about mentioning the release of water of different ages also as a function of the catchment's wet/dry conditions?*

We can certainly add those details. Again, due to heterogeneities, it would be difficult to do this in a catchment of this size as the stores of water are likely to differ spatially and there may also be differences in the timing that the stores (e.g., the soils or perched riparian groundwater) in different parts of the catchment become active.

**We have also mentioned this (lines 49-51); again, it is not something that we can address with our data and we have kept the discussion general.**

*Line 86: following up the comment of Anonymous Referee #1, quoted below "Since most earlier studies used monthly data with LPMs, I would not say that subweekly data are required when using attenuation of the stable isotope signal", I also suggest to reformulate the sentence, and stating why you say that sub-weekly or, more generally, high frequency tracer data are commonly needed. For example, it can be said that high-frequency and long-term tracer data are generally recommended to appropriately describe fast catchment-scale hydrological behaviors and the tail of the TTDs, respectively. See:*

*1. Kirchner, J. W., Feng, X., Neal, C., and Robson, A. J.: The fine structure of water-quality dynamics: the (high-frequency) wave of the future, Hydrol. Process., 18, 1353–1359, https://doi.org/10.1002/hyp.5537, 2004,*

*2. von Freyberg, J., Studer, B., and Kirchner, J. W.: A lab in the field: high-frequency analysis of water quality and stable isotopes in stream water and precipitation, Hydrol. Earth Syst. Sci., 21, 1721–1739, https://doi.org/10.5194/hess-21-1721-2017, 2017.*

This was discussed above in relation to the comments of Reviewer #1. It is true that MTTs have been estimated with less frequent data. However, as those references indicate, using more frequent data would allow for TTDs to be estimated and would also allow a better understanding of when and how different stores of water become activated. Our approach in this study to use $^3$H was that it allows MTTs to be estimated in relatively large rivers without the need for time-series $^{18}$O or Cl measurements that are not available in this catchment. We realise that this approach misses some of the details of the processes, but it does provide very valuable information on catchment functioning.

**As mentioned above, we have modified this material (lines 85-89).**

*Line 334-336: 'Overall, the major ion geochemistry of the groundwater, stream water from the different flow conditions, pool water and, NRW are similar'. What about EC? Differences in*

*EC between stream water (2430-15,330 μS cm-1) and near-river water (1035 to 6080 μS cm-1) during zero-flow period are significant, and you could explain why.*

In the upper Wimmera River, regional groundwater has the highest EC values and relatively high values of EC were also recorded in pool waters. On the contrary, stream water and near-river water (NRW) are less saline as they have contributions from fresh and young water stores. The difference in EC values between pool water and near-river water is mainly caused by evaporation in the pools. Additionally, the description in the previous version was probably confusing readers as 2430-15,330 μS cm$^{-1}$ was the range for pool water (we described it as stream water at zero flows).

**We have clarified this sentence (lines 334-336).**

*Line 416-418: 'The variation in ³H activities with δ²H (Fig. 6a) and TDS concentrations (Fig. 6b) most likely reflects the mixing between older regional groundwater and younger evaporated stream water'. It is not clear to me why you have drawn these conclusions. Could you explain better?*

This was also discussed above in relation to the comments of Reviewer #1. This explanation explains the available data (including the δ$^{18}$O and TDS). The sample with the highest δ$^2$H records evaporation (as implied by Fig. 4) and the samples with the lowest $^3$H look to have a component of older saline groundwater. As noted above, there may be a difference between the pools with larger groundwater contributions than those dominantly fed by surface water. **As mentioned above, we have added this explanation (lines 420-428).**

---

## Author Response (AR2)

Dear Editor,

We have addressed those minor comments as outlined below (line numbers refer to untracked version).

*Could you enlarge the precipitation bars in Fig. 2a, as they are hard to read (also precipitation is most often displayed as top-down bars). I think for the interpretation of your data it would be good to provide at least ETa data (in Fig. 2a).*

We have enlarged the precipitation bars and also made them top-down. Additionally, we have added ETa data to Fig. 2a and reported the trend in ET values in the text (lines 198-200).

*I was wondering if you could provide zoomed-in versions of the data clusters in Fig. 3, as the point clouds are hard to disentangle.*

We have added zoomed-in versions to Fig. 3 and have reduced the symbol size on this figure, which also improves clarity.

*The axis labels of Figs 5, 6, 8, 9 are quite small, please enlarge those.*

The axis text has been enlarged.

*Please provide an example of "future stresses" in line 593.*

'Future stress' has been omitted. We discuss the specific stresses (short-term variations and land use changes) on lines 578-579.

*I would suggest moving the section l. 611 ff. to the top of the conclusion section.*

We have moved the sentence to the start of the Conclusion (lines 569-571) and reworded a few of the other sentences so that it is clearer where we are discussing general issues.